# MoEs Are Stronger than You Think: Hyper-Parallel Inference Scaling with RoE

## Abstract

The generation quality of large language models (LLMs) is often improved by utilizing inference-time sequence-level scaling methods (e.g., Chain-of-Thought). We introduce *hyper-parallel scaling*, a complementary framework that improves prediction quality at the token level. Hyper-parallel scaling computes and aggregates multiple output proposals for a single token from the model. We implement this concept in Mixture-of-Experts (MoE) models, which we refer to as Roster of Experts (RoE). RoE is a training-free inference algorithm that turns a single MoE into a dynamic ensemble of MoEs. RoE injects controlled stochasticity into the expert routing mechanism, enabling it to sample multiple diverse experts for each token and aggregate their outputs for a more accurate final prediction. To overcome the computational cost, we introduce an efficient batching strategy and a specialized KV-caching mechanism that minimizes compute and memory overhead. For example, RoE enables a 7B MoE model to match the performance of a 10.5B MoE model while using 30% less compute for inference. These gains are achieved without any fine-tuning of model parameters.

## 1    Introduction

Extensive data and substantial computational resources have fueled recent advancements in language models. While the simplest method for generating responses is greedy decoding, the quality of model outputs often requires enhancement at inference time. A growing line of work in this area focuses on test-time scaling, which aims to improve the performance of the sequence generation process. Existing test-time scaling approaches typically fall into two orthogonal categories: sequential scaling, where the model produces longer, more structured outputs (e.g., Chain-of-Thought (Wei et al., 2022)); and parallel scaling, where multiple independent sequences are generated and then aggregated (e.g., self-consistency (Wang et al., 2022)). The general notion of these categories is marked as "Sequential Scaling" and "Parallel Scaling" in Figure 1.

In this paper, we pose an orthogonal question: Can we improve a model's intrinsic next-token prediction capability by allocating more computation at inference time? In other words, can we increase the model's internal compute during inference to enhance the quality of every generated token? We refer to this new paradigm as *hyper-parallel scaling*, as shown in Figure 1. This approach improves generation quality even under the simplest decoding strategy, greedy decoding. To isolate the gains attributable to hyper-parallel scaling, we focus our experiments on evaluating greedy decoding quality throughout the paper.

Hyper-parallel scaling aims to unlock a model's full potential by increasing the computation allocated to each token at inference time. One way to realize this idea is by introducing controlled variation within each transformer block (Shelmanov et al., 2021) and recomputing the layer output multiple times. Another approach is to reuse each layer repeatedly in a recurrent manner, thereby increasing computation without adding parameters (Lin et al., 2022). While many variants are possible, we focus on sparsely activated Mixture-of-Experts (MoE) models, which provide an ideal architecture for implementing this concept.

---

*Work done during an internship at Apple.
†Corresponding authors: szibakhshshabgahi@ucsd.edu, m_samraghrazlighi@apple.com

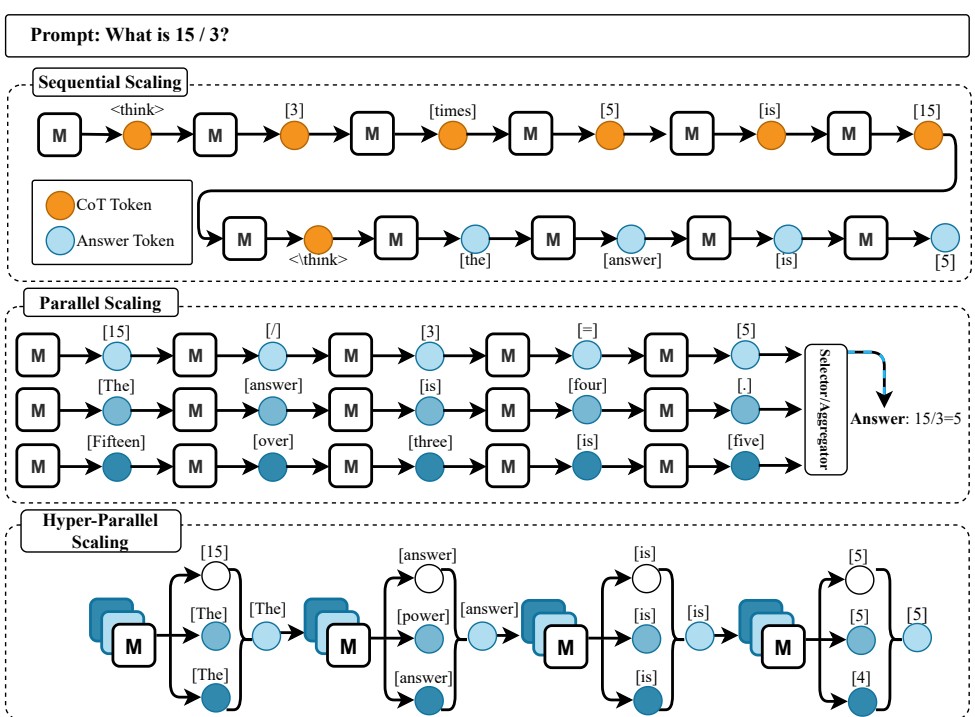

Figure 1: A categorization of inference-time scaling strategies. **(I) Sequential Scaling:** Enhancing performance by generating longer, structured outputs like a chain of thought (Wei et al., 2022). **(II) Parallel Scaling:** Generating multiple token sequences and aggregating them, as in Self-Consistency (Wang et al., 2022). **(III) Hyper-Parallel Scaling:** A novel paradigm, instantiated by RoE, that aggregates results from diverse internal computation paths on a per-token basis.

Mixture of experts (MoE) models have become a leading solution for frontier large language models (Shazeer et al., 2017; Comanici et al., 2025; Dai et al., 2024). Since they activate only a fraction of their parameters per forward pass, they naturally raise the central question of hyper-parallel scaling: can the inactive experts be leveraged at inference time to boost performance? Simply increasing the number of active experts does not work, as models are not trained to aggregate information from larger expert sets. To address this, we propose Roster of Experts (RoE), a training-free inference technique that treats a single MoE as a dynamic ensemble. RoE adds controlled stochasticity into the router's expert selection, runs multiple stochastic forward passes per token, and aggregates the resulting logits into a single, higher-quality prediction, all without model fine-tuning.

As is evident, a naive implementation of RoE would incur substantial redundant computation. We address this by exploiting the overlap across forward passes and merging them into a single batched call to the LLM. Furthermore, we introduce a specialized caching mechanism to reduce the KV-cache size required for RoE generation. In short, the contributions in this work are as follows:

- We introduce hyper-parallel scaling, a novel inference paradigm that allocates additional compute at test time to diversify a model's internal computations, thereby improving the quality of each token prediction.

- We propose Roster of Experts (RoE), a training-free approach to hyper-parallel scaling in MoE models that ensembles diverse computational paths. RoE leverages Gumbel-Top-K routing to inject controlled stochasticity into expert selection and introduces execution and KV-cache optimizations for efficient inference.

- We demonstrate the superior efficiency of RoE compared to conventional model scaling. For instance, we demonstrate that RoE can enhance the OlMoE-7B model (Muennighoff et al., 2024) to achieve the performance of a 10.5B model, with a 30% latency decrease compared to its larger counterpart. The enhancement requires no model finetuning.

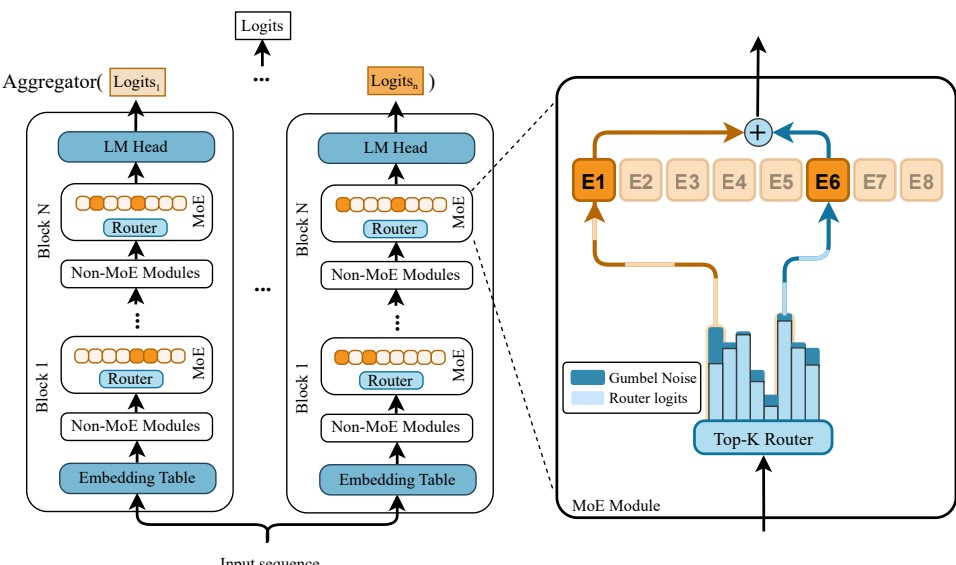

Figure 2: An illustration of the Roster of Experts (RoE) method. **Left:** For a single input, $n$ distinct experts are sampled by adding stochasticity to the expert routing at each MoE layer, and the resulting output logits are aggregated to form the final prediction. **Right:** A closer view of a single MoE layer shows $k = 2$ active experts (dark orange), where Gumbel noise (dark blue) is added to the router logits, and the top-$k$ experts are selected based on these modified logits.

## 2    RoE: Hyper Parallel Scaling of Mixture of Experts

Roster of Experts (RoE) enhances a pre-trained MoE model's performance by treating it as a dynamic ensemble. In designing RoE, we hypothesize that making controlled variations in routing still yields high-quality predictions. The rationale behind our claim is straightforward: during training, the model already encounters a wide range of expert combinations, so there is no reason not to exploit the same diversity at test time.

Given the aforementioned insight, we provide a high-level illustration of RoE in Figure 2. At each generation step, the MoE model generates multiple candidate output logits for a single input by sampling diverse expert selections from the model. These outputs are then aggregated to produce a single, more accurate prediction. This process relies on two key components: a stochastic routing mechanism to create diverse paths, and an efficient inference strategy to ensure practicality.

### 2.1    Gumbel-Top-K Routing for Path Diversity

Standard MoE models use deterministic top-k routing, where each token is routed to the $k$ experts with the highest router logits. To generate diverse computational paths, we introduce controlled stochasticity into this selection process using Gumbel-Top-K routing. Given the router logits $R \in \mathbb{R}^E$ for a token over $E$ experts, we perturb them with Gumbel noise before selecting the top-k experts. The indices of the selected experts are given by:

$$\text{Indices} = \text{TopK}(\mathbf{R} + \tau \cdot \mathbf{G}, k) \tag{1}$$

where $G$ is a vector of i.i.d. samples from the Gumbel(0, 1) distribution, and $\tau$ is a temperature parameter that controls the degree of stochasticity. When $\tau = 0$, this reduces to standard deterministic top-k routing. As $\tau$ increases, the selection becomes more random.

This method is a principled way to sample from the distribution implicitly defined by the router logits. The Gumbel-Max trick (Gumbel, 1954) establishes that adding Gumbel noise to logits before an $argmax$ operation is equivalent to sampling from the categorical distribution produced by applying a softmax to the logits. By extension, applying a TopK operation to Gumbel-perturbed

logits corresponds to sampling $k$ elements without replacement from the distribution. This ensures that experts with higher router logits remain more likely to be selected even after adding Gumbel noise, preventing the selection from drifting too far from the trained router's predictions. Figure 2 illustrates this mechanism.

## 2.2 Choosing the Gumbel Temperature

The Gumbel temperature, $\tau$, controls the degree of stochasticity in expert routing. We treat it as a layer-specific hyperparameter, defining a temperature vector $\boldsymbol{\tau} = \{\tau_i\}_{i \in \mathcal{L}_{MoE}}$, where $\mathcal{L}_{MoE}$ is the set of MoE layers. Setting a small value of $\tau$ keeps router selections nearly unchanged, reducing expert diversity per sample and making the next-token prediction closely match the underlying MoE. In contrast, an excessively large $\tau$ introduces too much randomness in expert selection, degrading prediction quality. Appendix 3.6 illustrates how task performance varies as the temperature increases. Selecting the optimal $\boldsymbol{\tau}$ presents a hyperparameter optimization problem, balancing the potential performance gain against the cost of the search. The primary challenge is the search space, which grows exponentially with the number of MoE layers, rendering an exhaustive search infeasible. A practical search strategy therefore requires an efficient optimization algorithm and a carefully chosen validation metric.

### 2.2.1 Optimization Metric

We consider two metrics for guiding the hyperparameter search: validation perplexity and task-specific accuracy.

**Validation Perplexity (PPL)** is computationally inexpensive, as it only requires a single forward pass over the validation set. However, PPL is an indirect measure of generative reasoning. A low PPL indicates that the model assigns a high probability to a ground-truth sequence, but does not guarantee that the model can generate a correct solution independently.

**Validation Accuracy** directly measures the model's ability to solve the task, making it a more faithful metric for generative performance. The main drawback is its high computational cost, as it requires generating a full solution for each validation example. This cost can make it prohibitive for large-scale hyperparameter searches.

### 2.2.2 Search Strategy

Given the cost of each evaluation, Bayesian optimization methods are well-suited for this task. In our experiments (Section 3), we employ the Tree-structured Parzen Estimator (TPE) Watanabe (2023) via the Optuna framework Akiba et al. (2019). To make the search tractable for models with many MoE layers, we introduce two heuristics to prune the vast search space, based on empirical observations:

**1. Apply RoE to middle layers only.** We hypothesize that the initial and final layers of a transformer are more sensitive to routing perturbations. The initial layers process raw token embeddings, while the final layers consolidate information for the output prediction. Our experiments show preference of the optimizer to reduce the stochasticity of initial and final layers. We therefore constrain the search by setting the temperature to zero ($\tau_i = 0$) for the first and last $k$ MoE layers, applying RoE only to the intermediate ones.

**2. Bound the temperature range.** We empirically observe that temperature values above 0.5 introduce excessive routing noise, which consistently harms model performance. Consequently, we restrict the search space for each non-zero temperature to the range $[0, 0.5]$.

## 2.3 Efficient RoE Inference

A naive implementation of RoE, which performs $n$ independent forward passes, would incur a prohibitive n-fold increase in computation. We introduce two optimization techniques that drastically reduce this overhead.

First, we take advantage of the batched parallel processing capabilities of modern accelerators, such as GPUs. The latency of a forward pass grows sub-linearly with the batch size due to hardware-level parallelization. By processing the $n$ samples for a single token generation step as a batch, we can significantly reduce the wall-clock time compared to $n$ sequential runs.

We demonstrate in Section 3.3 that Key-Value (KV) caching significantly reduces the computational overhead of sequence generation with RoE. However, batched inference alone does not solve the significant memory overhead introduced by the KV-cache in autoregressive decoding. Since the $n$ samples follow different computational paths, their hidden states diverge. A naive implementation would require maintaining $n$ separate KV caches, one for each sample's unique history. This scales both memory and computation linearly with $n$, quickly becoming intractable for long sequences.

To address this, we introduce a novel caching strategy named **Clean Cache**1. Our key insight, supported by preliminary experiments showing no significant performance drop compared to maintaining separate histories, is that sufficient output diversity can be achieved by applying stochastic routing only for the current token2. We implement this efficiently by setting the routing temperature $\tau$ for the first sample (batch index 0) to zero to establish a "clean" path3. We then reuse this single deterministic KV cache across all other samples in the batch4. Consequently, the memory footprint remains identical to that of a single sample 5, localizing the additional cost of RoE entirely to the current batched forward pass.

## 3 EXPERIMENTS

### 3.1 EXPERIMENTAL SETUP

**Models and Benchmarks.** We evaluate RoE on three instruction-tuned MoE models: OLMoE-1B-7B-Instruct (Muennighoff et al., 2024), Mixtral-8x7B-Instruct-v0.1 (Jiang et al., 2024), and GPT-OSS-20B (OpenAI et al., 2025). Our evaluation spans three domains: mathematical reasoning (GSM8K (Cobbe et al., 2021), SVAMP (Patel et al., 2021), AddSub (Hosseini et al., 2014), SingleEQ (Koncel-Kedziorski et al., 2015), MultiArith (Imani et al., 2023)), commonsense reasoning (ARC-Easy, ARC-Challenge (Clark et al., 2018), OpenBookQA (Mihaylov et al., 2018), Social-I-QA (Sap et al., 2019), Hellaswag (Zellers et al., 2019)), and code generation (HumanEval (Chen et al., 2021), HumanEvalPlus (Liu et al., 2023)).

**Baseline and Evaluation Strategy.** Our primary goal is to isolate the performance gains directly attributable to RoE's modification of the model's internal computational pathways. Unlike test-time methods that treat the model as a black box and ensemble outputs from multiple generation runs (e.g., self-consistency), RoE enhances the model's backbone directly. These two classes of methods are orthogonal and can be used in conjunction. Therefore, to ensure a clean and direct measurement of RoE's impact, we use the standard, unmodified MoE inference as our sole baseline. We employ greedy decoding for all experiments, ensuring that any observed improvements stem from RoE itself, not from interactions with complex decoding strategies.

**RoE Configuration and Tuning.** We tune the per-layer noise temperatures for RoE on each task's validation set using the Tree-structured Parzen Estimator (TPE) (Watanabe, 2023) implemented in Optuna (Akiba et al., 2019). All reported results are evaluated on the **test sets** after tuning, ensuring no data contamination from the **validation sets**. For computational efficiency, we optimize validation perplexity for math tasks and validation accuracy for commonsense and code tasks. We employ our 'clean-cache' implementation (Section 2) for OLMoE and the standard cache for other models to demonstrate RoE's effectiveness under both caching strategies. The tuning budget and key RoE hyperparameters are summarized in the appendix (Table 1, while the complete per-layer temperature profiles are visualized in the appendix (Figure 11).

**Implementation Details.** Experiments use NVIDIA A100 80GB GPUs. RoE predictions are aggregated by probability averaging. For commonsense tasks, we follow Hu et al. (2023) and select the answer with the highest log-probability. Results are averaged over 5 seeds. Inference latency is measured as the wall-clock time for generating 128 tokens. We report exact-match accuracy for all tasks and pass@1 additionally for code tasks.

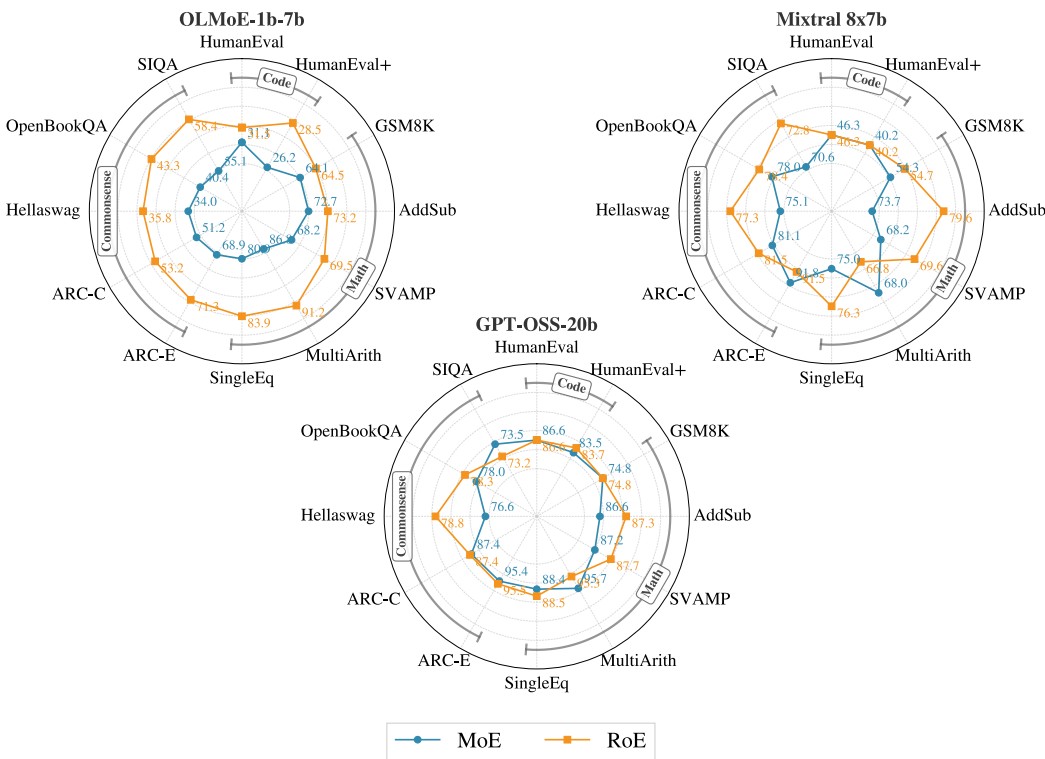

Figure 3: Performance comparison of base MoE models and RoE on five mathematical, five commonsense, and two code benchmarks. Accuracy is measured by exact match (except for HE and HE+, which use pass@1). Results are averaged over five random seeds. Axes are normalized to $(\min - 1, \max + 1)$ for visualization.

## 3.2 MAIN RESULTS

Figure 3 presents the performance of RoE applied to the three MoE models across our suite of benchmarks. As is evident, RoE consistently improves performance across nearly all tasks and model scales, demonstrating its broad effectiveness as a post-hoc enhancement strategy.

The performance gains are most pronounced for OLMoE, the smallest and weakest base model, where RoE substantially lifts accuracy across all benchmark categories. We hypothesize that by diversifying computational paths, RoE unlocks latent capabilities inaccessible to a single, static routing configuration. This suggests the diversification is most impactful for models with more initial headroom for improvement.

However, the improvements are not uniform. For math benchmarks, we optimized RoE's hyperparameters for validation set perplexity. While RoE consistently lowered perplexity (see Figure 9 in the appendix), this did not always translate to higher generative accuracy. For instance, on Multi-Arith, RoE did not improve the accuracy of Mixtral or GPT-OSS. This result underscores the known disconnect between perplexity and generative reasoning performance, suggesting that tuning RoE directly on task-specific metrics could yield further gains where computationally feasible.

Finally, we observe a ceiling effect: as a model's baseline performance on a benchmark approaches saturation, the potential for improvement diminishes. For instance, the GPT-OSS model already achieves 95.7% on MultiArith, leaving minimal room for any method to provide substantial gains.

A key advantage of RoE is its applicability to open-ended generation, where parallel scaling methods, like majority voting over complete outputs, are infeasible. By aggregating logits at each token-generation step, RoE enhances the performance of open-ended tasks such as code generation, as evident in Figure 3.

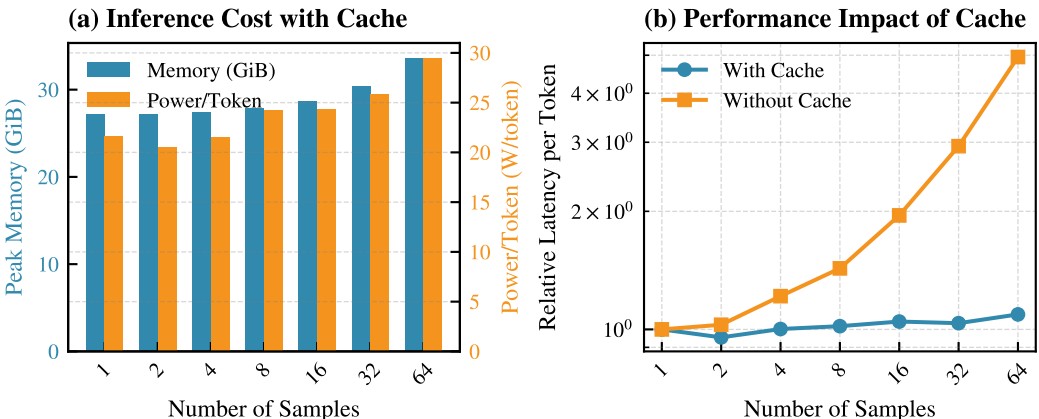

Figure 4: Impact of caching on RoE performance. **(a)** Resource usage with caching enabled. Peak memory (blue, left axis) and power per token (orange, right axis) show a modest increase as the sample count grows. **(b)** Latency comparison with and without caching. Without caching, latency per token rises exponentially, highlighting the necessity of caching for scalable RoE inference.

### 3.3 COMPUTATIONAL OVERHEAD

We evaluate the computational overhead of RoE in terms of GPU memory footprint, power usage, and throughput (tokens/second). Our benchmark uses the OLMoE model to generate solutions for the first 100 problems from the GSM8k benchmark (Cobbe et al., 2021) on a single NVIDIA A100 GPU. To isolate the overhead of our method and ensure a fair comparison, we set the generation temperature and $\tau$ to 0, which produces identical output sequences across all runs.

Figure 4(a) illustrates the relative increase in inference cost for RoE with our caching mechanism. The resource requirements grow modestly with the number of samples. For instance, using 64 samples increases the memory footprint by only 12% and power consumption by 20%. These results show that the additional computational cost is manageable and not prohibitive.

Figure 4(b) underscores the necessity of our caching scheme. By comparing RoE with and without caching, we observe that our approach drastically mitigates the increase in power consumption and latency. Without caching, the cost of sampling multiple paths would be impractical. These findings confirm that RoE offers a practical trade-off, enhancing model quality for a moderate and controllable increase in computational demand.

### 3.4 EFFICIENCY ANALYSIS: RoE vs. MODEL SCALING

RoE enhances a model's performance, making it comparable to a larger version of the base model. To quantify this improvement, we measure the *equivalent model size* that a standard MoE would need to match the performance of RoE for a given number of samples $K$. This allows us to directly compare the efficiency of RoE against the cost of simply using a larger MoE. We estimate this equivalent size by applying the scaling laws from Kaplan et al. (2020), using test perplexity on the WikiText-103 dataset (Merity et al., 2016) as the performance metric. While we acknowledge the imperfect correlation between perplexity and downstream performance, this established framework provides the most practical resource estimation benchmark in the absence of larger trained OLMoE variants. For each sample size $K$, we tune the RoE routing temperatures across all layers by optimizing perplexity on the WikiText-103 validation set over 50 trials with a TPE optimizer. We then measure the test perplexity of the best-performing temperature setting on the validation set. As shown in Figure 5(a), the effective model size grows monotonically with $K$, although the gains diminish as the sample size increases and the set of viable expert pathways saturates.

Figure 5(b) compares the computational cost of RoE (blue curve) with the cost of using a larger, equivalently performing standard MoE model (orange curve). The results show that RoE is substantially more efficient in terms of both latency and memory. For instance, applying RoE with $K = 32$

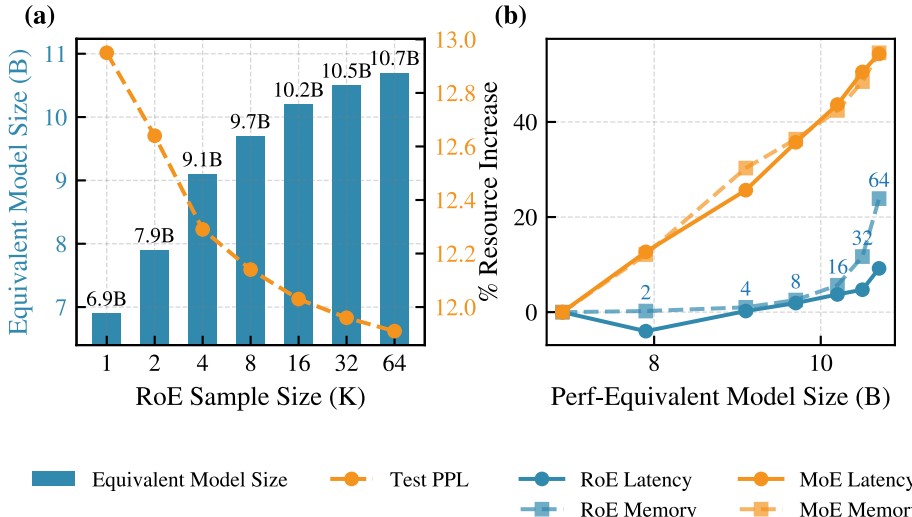

Figure 5: Performance and efficiency analysis of RoE. **(a)** The performance of RoE applied to OLMoE-7B, measured in terms of an equivalent standard MoE model size. Performance is evaluated using perplexity on the WikiText-103 test set. **(b)** Comparison of the relative increase in latency and memory for RoE (blue) versus scaling up to an equivalently performing MoE model (orange). The numbers on the blue curve indicate the RoE sample size $K$.

to the OLMoE-7B model yields performance comparable to that of a 10.5B OLMoE model. Notably, this configuration reduces memory overhead by 25% and per-token latency by 30% relative to the larger model, underscoring the efficiency of RoE as an alternative to conventional model scaling.

## 3.5 RoE an Orthogonal Scaling Direction

We investigate the interaction between RoE and sequence-level aggregation methods, specifically Self-Consistency (SC) (Wang et al., 2022). While RoE (Hyper-Parallel scaling) enhances the model's intrinsic generation capabilities by diversifying internal computational paths, SC (Parallel scaling) improves performance by selecting the most consistent output across multiple independent generations. However, a key limitation of SC is its reliance on tasks where a distinct final answer can be extracted and aggregated. In contrast, RoE operates directly at the token-level, making it applicable to any generative task, including open-ended generation where answer aggregation is infeasible. Despite these functional differences, because they target distinct stages of the inference process, they can be complementary. To demonstrate this complementary relationship, we conducted an experiment on the Sin-

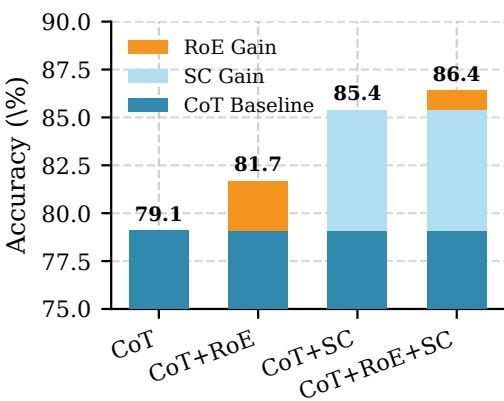

Figure 6: Ablation study showing the additive gains of RoE and Self-Consistency over the CoT baseline.

gleEQ Koncel-Kedziorski et al. (2015) benchmark comparing RoE ($K = 32$) against Self-Consistency ($N = 5$) and their combination. As illustrated in Figure 6, both methods individually outperform the greedy Chain-of-Thought (CoT) baseline (79.1%). RoE achieves 81.7% and SC achieves 85.4% in isolation. Crucially, the combined approach (**SC + RoE**) yields a substantial improvement, reaching an accuracy of **86.4%**. This result confirms that RoE enhances the model's

fundamental token-prediction quality, which in turn provides a stronger probability distribution for sequence-level techniques like SC to exploit.

### 3.6 IMPACT OF ROUTING TEMPERATURE

The routing temperature is a key hyperparameter in RoE, governing the diversity of sampled expert paths. To understand its effect, we conduct a sensitivity analysis where we apply a uniform temperature across all MoE layers and sweep its value in increments of 0.05.

As shown in Figure 7, performance consistently follows a concave trend: accuracy improves as the temperature increases from zero, peaks at an optimal value, and then declines. This decline occurs because excessively high temperatures introduce too much noise into the routing decisions, leading to the selection of less relevant experts and degrading the final prediction quality. Furthermore, we observe that sparsity levels influence hyperparameter sensitivity rather than total gain: highly sparse models (e.g., GPT-OSS) are more sensitive to noise than less sparse ones (e.g., Mixtral).

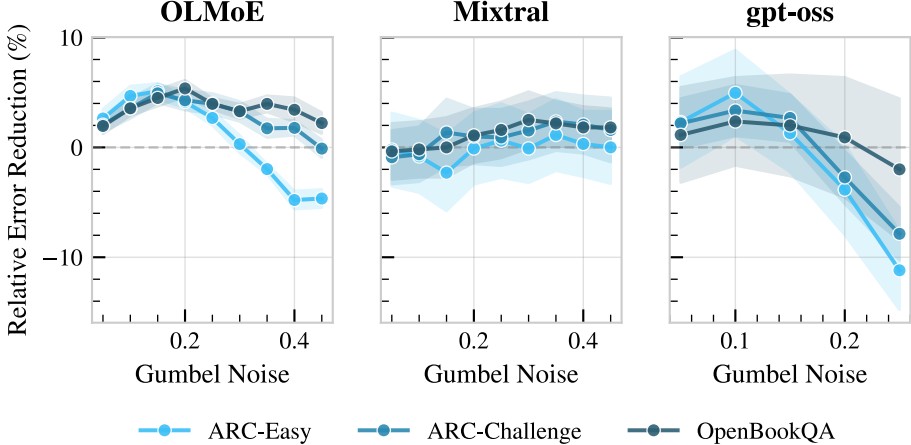

Figure 7: Impact of routing temperature on task performance. We apply a uniform temperature across all MoE layers and observe a concave relationship where performance peaks at a task-specific optimal value. Excessively high temperatures degrade performance by introducing noise into the expert selection process.

## 4 RELATED WORK

Our work, Roster of Experts (RoE), introduces a method for trading additional inference-time compute for improved model performance. Several strategies improve model performance by increasing computation at inference time. Inspired by Mirtaheri et al. (2025) we categorize them into three paradigms, as illustrated in Figure 1.

**Sequential Scaling:** Sequential scaling methods improve reasoning quality by prompting the model to generate longer, more structured intermediate steps. The seminal work in this area is Chain-of-Thought (CoT) prompting (Wei et al., 2022; Nye et al., 2021), which elicits step-by-step reasoning. In this category, we ask the model to think, question it self and do a step by step analysis (OpenAI, 2025; Kojima et al., 2022). This approach has been extended by a large body of subsequent work that explores more complex reasoning structures, such as Tree-of-Thoughts (Yao et al., 2023), at the cost of increased sequential generation steps. The language model needs to be trained with CoT prompts to be able to utilize this capability during inference.

**Parallel Scaling:** Parallel scaling generates multiple independent outputs for the same prompt and aggregates them to produce a final, more robust answer (Brown et al., 2024). The simplest technique to mention for this category is beam search Freitag & Al-Onaizan (2017), where multiple sequences of tokens are evolved at the inference time and the best sequence is chosen as the response. Another

prominent example is Self-Consistency (SC) (Wang et al., 2022), which samples diverse reasoning paths using stochastic decoding (e.g., non-zero temperature) and selects the most frequent answer via majority vote. While highly effective, the standard voting mechanism limits SC to tasks with easily verifiable answers, such as math or multiple-choice questions. Recent work aims to extend this paradigm to open-ended generation tasks by developing more sophisticated aggregation strategies that do not rely on a single, extractable answer and instead aggregate on a sequence level (Chen et al., 2023; Taubenfeld et al., 2025; Wang et al., 2024).

**Hyper-Parallel Scaling:** We introduce *hyper-parallel scaling* as a distinct, third paradigm. Unlike sequential and parallel methods that scale the *output generation process*, hyper-parallel scaling diversifies the *internal computation paths* within the model for a single token prediction. RoE actualizes this concept by treating a single MoE model as an ensemble of subnetworks. By sampling different sets of active experts for each forward pass, RoE aggregates multiple internal predictions to produce a single, higher-quality output distribution for the next token. This approach is orthogonal to and can be combined with both sequential and parallel scaling.

A related direction was explored by Geiping et al. (2025), who designed a special recurrent architecture allowing for variable-depth inference. However, their method requires a model to be specifically pretrained for this capability. In contrast, RoE is a training-free strategy that can be applied post-hoc to any pretrained MoE model, offering a practical way to unlock enhanced performance from existing artifacts dynamically, and on demand.

## 5 CONCLUSION

We introduced hyper-parallel scaling, a novel inference paradigm for improving model quality that is orthogonal to existing sequence-level approaches. Instead of generating multiple sequences, this paradigm enhances a model's intrinsic next-token prediction by diversifying its internal computation. We presented Roster of Experts (RoE), a training-free realization of this concept for Mixture-of-Experts (MoE) models. RoE treats a single MoE model as a dynamic ensemble, leveraging controlled stochasticity in the expert routing to activate diverse experts and aggregate their outputs into a more robust prediction. Crucially, we demonstrated that through efficient batching and caching, the computational overhead of this method is minimal. Our results show that RoE enables a model to achieve the performance of a substantially larger counterpart with only a minor increase in inference cost. This provides practitioners with a powerful tool to dynamically trade inference-time compute for higher quality from a single, pre-trained model.

## 6 FUTURE WORK

Our work on Roster of Experts (RoE) opens several promising avenues for future research. We highlight three key directions:

- **Generalizing Hyper-Parallel Scaling:** A key direction is to extend hyper-parallel scaling beyond MoE models. For instance, in dense architectures, inference-time dropout could be employed to stochastically mask activations, inducing diverse high-quality outputs for aggregation. Unlike sequence-level scaling methods that are specific to auto-regressive tasks, hyper-parallel scaling is domain-agnostic. This is particularly relevant for modalities like vision, audio, and video, where test-time performance scaling is largely unexplored.

- **Advanced Noise Injection:** Developing more sophisticated noise schemes, such as adaptive or input-conditioned noise, to achieve more effective diversification of expert selection and further improve performance. Another avenue of study would look into stochasticity mechanisms beyond Gumbel noise.

- **Adaptive Computation:** Investigating strategies to dynamically adjust RoE's computational budget, for instance by varying the ensemble size based on token difficulty, to optimize the performance-cost trade-off.

- **RoE-Aware Training:** Incorporating stochastic routing into the pre-training or fine-tuning process to create models explicitly optimized for RoE, potentially yielding greater gains in efficiency and quality.

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
