# A EXPERIMENT SETUP DETAILS

Table 1: Experiment setup for model tuning across different task categories. For each category, we specify the tuning configuration and model-specific hyperparameters. $N$ denotes the number of RoE samples, $T$ is the maximum temperature in tuning, and $L$ refers to the number of initial and final skipped layers. Sample denotes the number of validation set samples, Trial is the number of optimization trials. PPL denotes if perplexity was used as the optimization objective.

| Category | Task | Sample/Trial | PPL | OLMoE | | | Mixtral | | | GPT-OSS | | |
|---|---|---|---|---|---|---|---|---|---|---|---|---|
| | | | | $N$ | $T$ | $L$ | $N$ | $T$ | $L$ | $N$ | $T$ | $L$ |
| Math | GSM8K AddSub SVAMP MultiArith SingleEq | 100/50 | ✓ | 32 | 0.5 | 1 | 64 | 0.25 | 5 | 64 | 0.2 | 5 |
| Common | SiQA OBQA Hellaswag ARC-E ARC-C | 300/100 | ✗ | 32 | 0.5 | 3 | 64 | 0.3 | 3 | 64 | 0.2 | 5 |
| Code | Humaneval Humaneval+) | 50/50 | ✗ | 32 | 0.5 | 1 | 64 | 0.25 | 5 | 64 | 0.2 | 5 |

## B  IMPACT OF ROUTING TEMPERATURE

The routing temperature is a key hyperparameter in RoE, governing the diversity of sampled expert paths. To understand its effect, we conduct a sensitivity analysis where we apply a uniform temperature across all MoE layers and sweep its value in increments of 0.05.

As shown in Figure 6, performance consistently follows a concave trend: accuracy improves as the temperature increases from zero, peaks at an optimal value, and then declines. This decline occurs because excessively high temperatures introduce too much noise into the routing decisions, leading to the selection of less relevant experts and degrading the final prediction quality. Crucially, we observe that the optimal temperature is task-specific, which underscores the importance of tuning this hyperparameter for each downstream application to maximize performance gains.

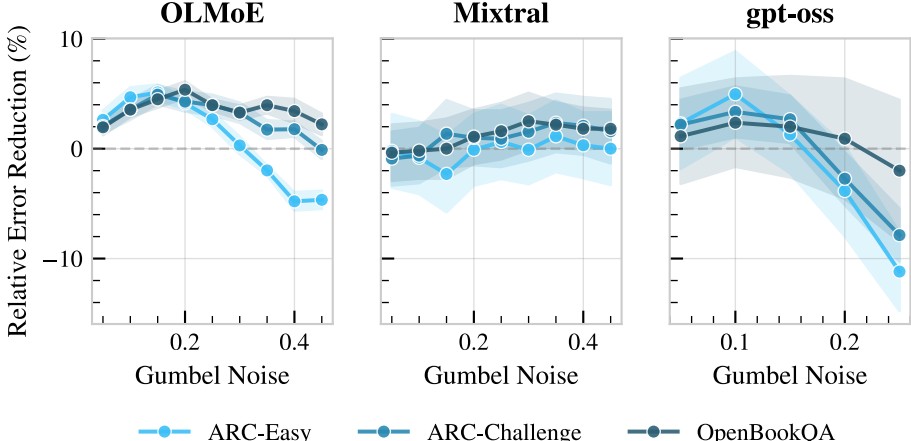

Figure 6: Impact of routing temperature on task performance. We apply a uniform temperature across all MoE layers and observe a concave relationship where performance peaks at a task-specific optimal value. Excessively high temperatures degrade performance by introducing noise into the expert selection process.

## C  TUNING HISTORY AND RESULTS HEATMAP

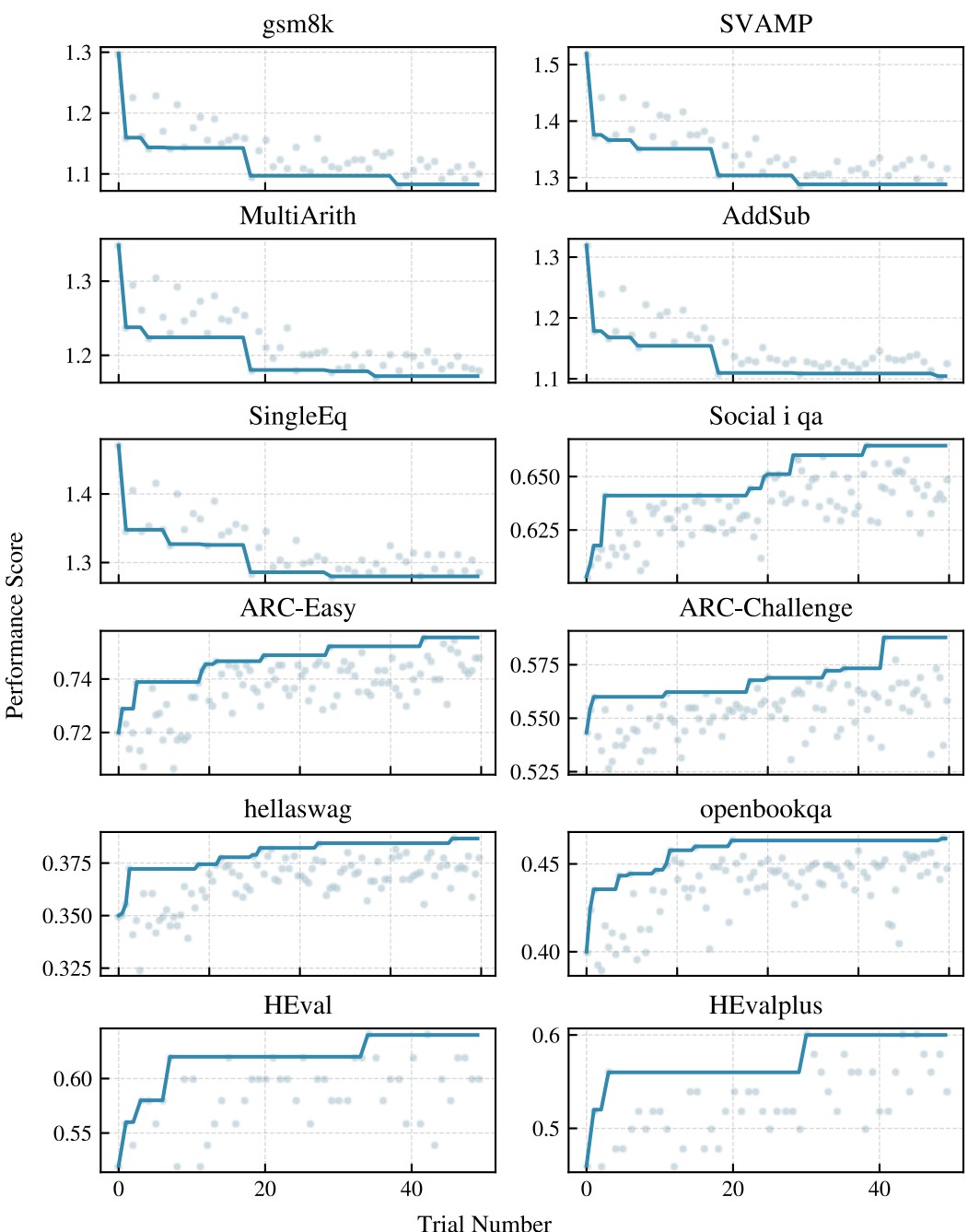

Figure 7: Optimization history of OLMoE on the benchmarks on the validation set. Test set evaluation results are provided in Figure 3

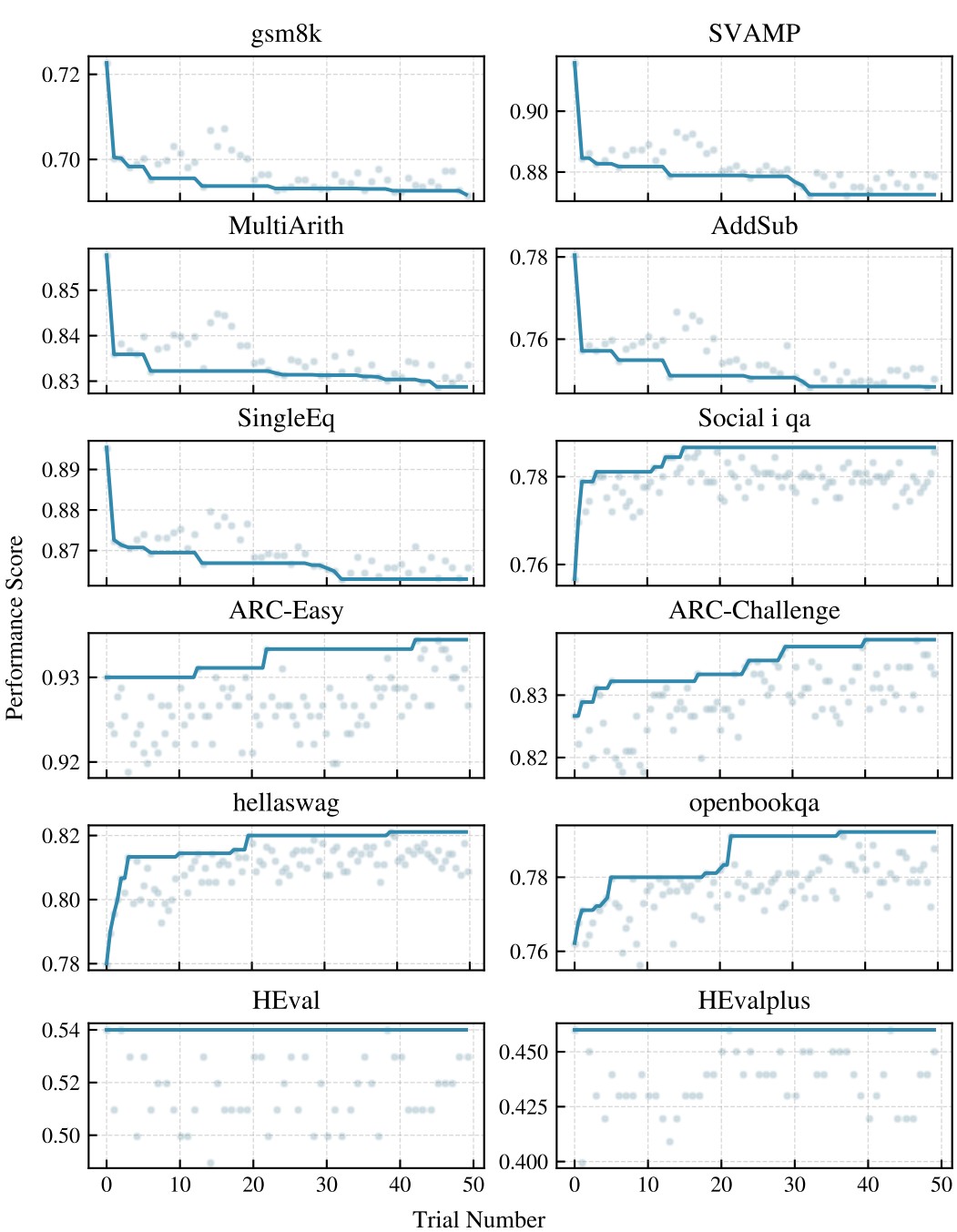

Figure 8: Optimization history of Mixtral on the benchmarks on the validation set. Test set evaluation results are provided in Figure 3

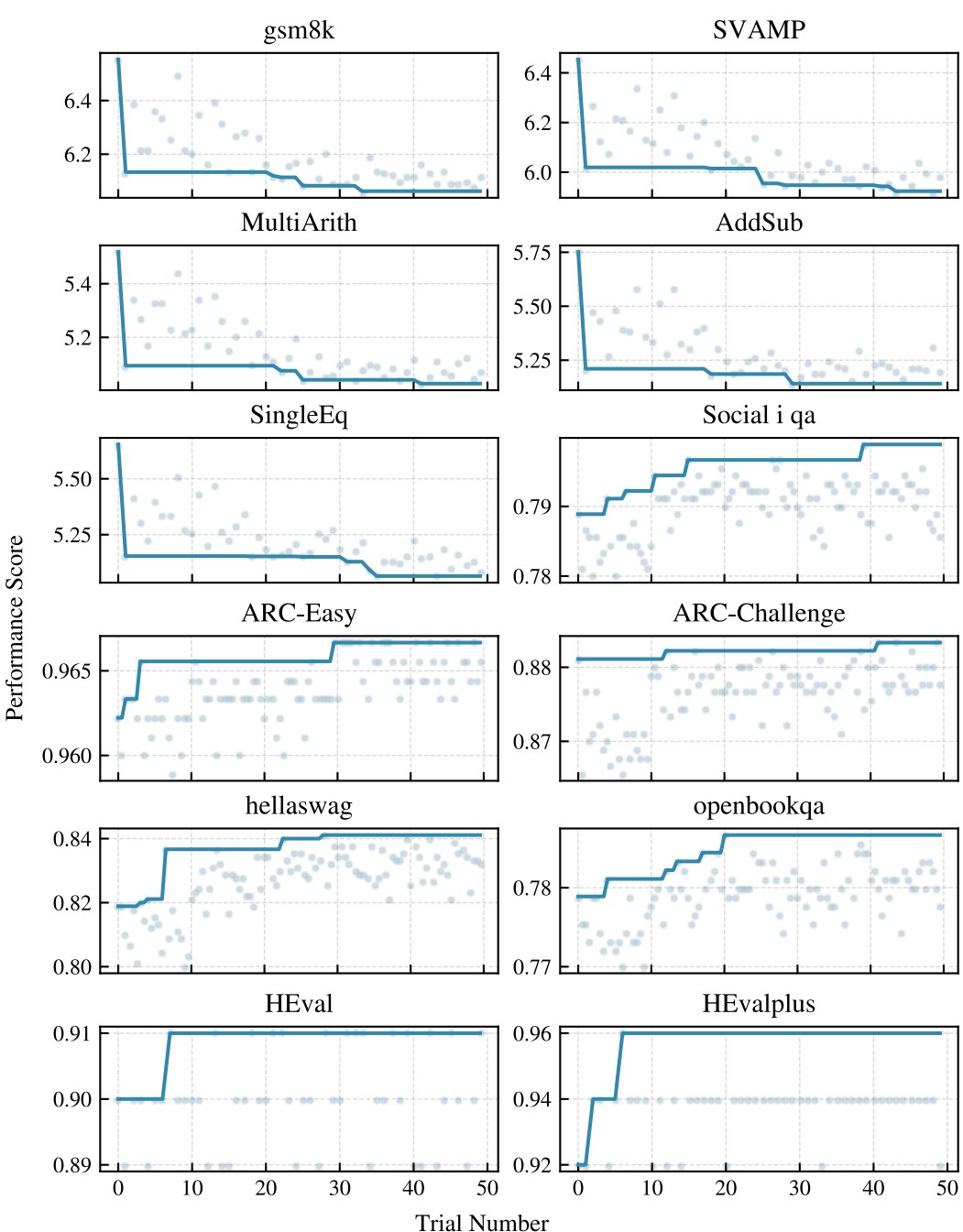

Figure 9: Optimization history of GPT-OSS on the benchmarks on the validation set. Test set evaluation results are provided in Figure 3

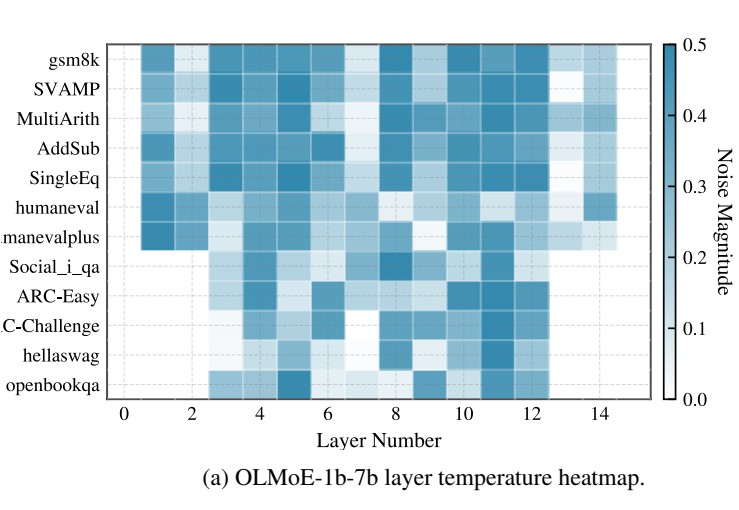

(a) OLMoE-1b-7b layer temperature heatmap.

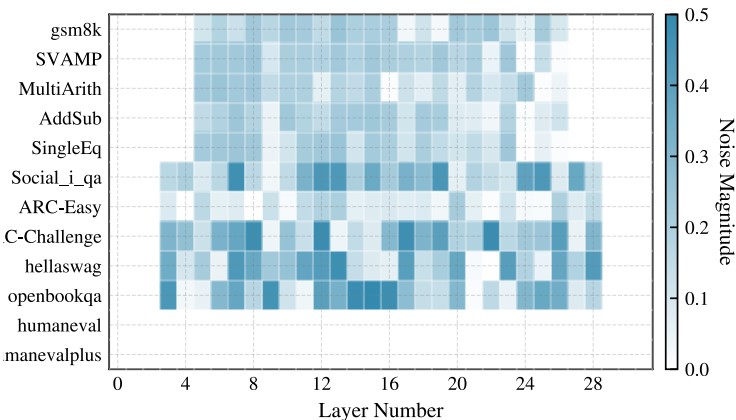

(b) Mixtral-7x8 layer temperature heatmap.

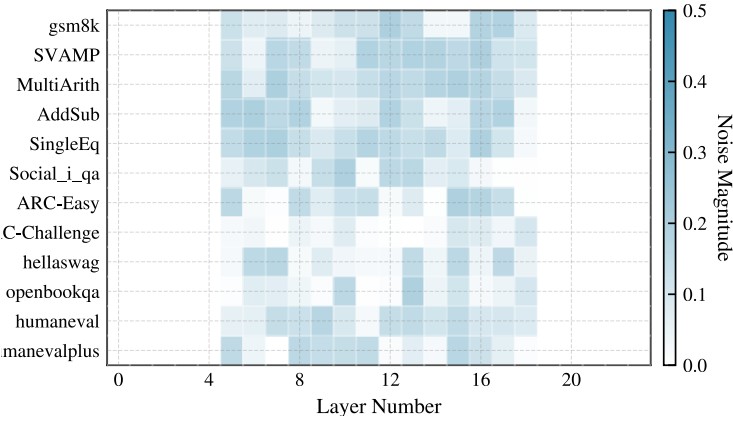

(c) GPT-OSS-20B layer temperature heatmap.

Figure 10: Heatmap of per layer temperature of OLMoE, Mixtral, and GPT-OSS after hyper-parameter tuning on different layers.