# OpenReview forum: "MoEs Are Stronger than You Think: Hyper-Parallel Inference Scaling with RoE"
_ICLR.cc/2026/Conference — Submitted to ICLR 2026_

### Official Review · Reviewer_rwhF · 2025-10-22

**Soundness:** 3
**Presentation:** 3
**Contribution:** 3
**Rating:** 2
**Confidence:** 3

**Summary:**

The authors introduce a training-free framework complementary to chain-of-thought scaling by enabling parallel inference in MoE via repeatedly sampling activated experts, before aggregating the outputs from multiple computational paths to produce a more accurate next-token prediction. Experiments on three different MoE models across mathematical and commonsense reasoning and code generation benchmarks demonstrate that the proposed RoE method consistently improves performance.

**Strengths:**

1. The paper is well written and well motivated. It complements existing inference scaling paradigms.

2. The empirical results are comprehensive and demonstrate a new axis for scaling.

**Weaknesses:**

1. The "equivalent model size" is a weak argument, as the authors point out the disconnect between perplexity and downstream task performance. While this remains a challenge in scaling law research, which is out of scope of this investigation, this nevertheless renders the argument less convincing. In addition, Figure 5 seems to suggest a diminishing return as the number of samples increases. What is the performance ceiling of RoE, and how does it compare to a vanilla best-of-N scaling?

2. The author presents clean cache as a major contribution, but only evaluated it on one of the three models tested without offering a clear explanation. This seems to greatly undermine the case the authors attempt to build.

3. The authors did not mention the cost associated with the Bayesian Optimization-based search. The authors present the layer-wise temperature as a parameter that requires careful tuning, but did not further discuss how difficult it is to obtain a good configuration nor how much time it takes.

4. MoE as a class of models also varies significantly in how sparse they are. More discussions on how sparsity affects performance improvements could be added to understand the tradeoff.

5. While the method is orthogonal to sequential scaling paradigms (such as CoT), iso-compute ablation studies are notably missing from the experiment sections. For instance, a simple baseline to compare against could be self-consistency.

**Questions:**

Please see my points above.

---

> ### Author Response · Authors · 2025-11-21
>
> We thank the reviewer for their thorough and insightful comments, and for finding the manuscript well-written, motivated, and supported by strong empirical results. We appreciate the opportunity to clarify the cost and implementation details.
>
> **“The equivalent model size is a weak argument...”**
> We agree that PPL implies limitations regarding downstream accuracy [Lines 185-188]. The purpose of the calculation is to compare RoE's resource overhead against the industry standard of "scaling up." In the absence of published larger OLMoE variants, scaling laws [1] provide the most practical framework to estimate if RoE's modest compute increase is more efficient than deploying a larger model. We will strengthen this discussion in Section 3.4.
>
> [1] Kaplan et al., *Scaling laws for neural language models*, arXiv 2020.
>
> **“Figure 5 seems to suggest a diminishing return... performance ceiling...”**
> Like most scaling mechanisms, RoE exhibits diminishing returns, saturating around $K=64$ in our experiments. This occurs because most viable computational pathways are eventually sampled, and increasing $K$ further primarily introduces redundancy. Figure 5 quantifies this trade-off to help practitioners choose an optimal $K$.
>
> **“How does it compare to a vanilla best-of-N (BoN) scaling?”**
> BoN generally assumes a verifier or reward model. RoE is a generative enhancement that does not require external verification. However, a BoN setup would benefit from using an RoE-enhanced model as the generator. We address this complementarity via the new Self-Consistency (SC) experiment below.
>
> **“Clean cache... only evaluated on one of the three models...”**
> We apologize for the confusion.
> * **Role:** Clean Cache is an efficiency optimization, not the central theoretical contribution (Hyper-Parallel Scaling).
> * **Empirical Justification:** Initial experiments on math tasks showed **no statistically significant difference** in accuracy between Clean and Regular Cache. This suggests local diversification at the current token is sufficient to realize performance gains.
> * **Clarification:** We intentionally used Clean Cache for OLMoE and Regular Cache for Mixtral/GPT-OSS to demonstrate that *both* strategies are feasible. We will explicitly clarify this in Section 2.3 and 3.1.
>
> **“Cost associated with Bayesian Optimization-based search.”**
> Valid point. The cost is defined by $N_{samples} * N_{trials} * T_{inference}$.
> * **Wall-clock (single NVIDIA A6000 for OLMoE):**
>     * Math Benchmarks: ~28 minutes total.
>     * Commonsense Benchmarks: ~2 hours 40 minutes total.
> This is a one-time "calibration" cost, significantly cheaper than fine-tuning. We will add these metrics to the Appendix.
>
> **“Discussions on how sparsity affects performance...”**
> RoE leverages "inactive but capable" experts. We observe that models with higher sparsity (e.g., GPT-OSS (4/128) > OLMoE (8/64) > Mixtral (2/8)) are more sensitive to temperature $\tau$. While overall performance improvements do not strictly correlate with sparsity, the *sensitivity* of the temperature response does (see Appendix Figure 6). We will add this insight to Revision, new Section 3.6.
>
> **“Iso-compute ablation studies... baseline to compare against could be self-consistency.”**
> We agree. RoE (Hyper-Parallel) improves intrinsic generation, while SC (Parallel) improves output selection. We performed the requested comparison on the SingleEQ benchmark, demonstrating that RoE and SC are complementary. For SC the setup of
>
> **Table 1: Performance Comparison on SingleEQ**
>
> | Method | Configuration | Accuracy |
> | :--- | :---: | :---: |
> | CoT (Baseline) | $K=1$ | 79.1% |
> | CoT + RoE | $K=32$ | 81.7% |
> | CoT + SC | $N=5$ | 85.4% |
> | **CoT + RoE + SC** | **SC ($N=5$) on RoE ($K=32$)** | **86.4%** |
>
> *We use $K=32$ samples for RoE and $N=5$ samples for Self-Consistency.*
>
> **Conclusion**
> We believe the inclusion of the SC+RoE experiment, the clarification regarding Clean Cache sufficiency, and the detailed breakdown of tuning costs address your major concerns. We hope these clarifications and the demonstrated efficiency gains justify a favorable reconsideration of your score.
>
> **List of Revisions:**
> * **Section 3.5 (Lines 407-430):** Added new experiment (RoE vs SC vs Combined) with Figure 6.
> * **Lines 227-229, 261-263:** Clarified Clean Cache scope and effect.
> * **Lines 443-445:** Added discussion on sparsity and temperature sensitivity.
> * **Line 373-375:** Added discussion on performance ceiling.
> * **Lines 376-406:** Strengthened discussion on model size arguments.

---

### Official Review · Reviewer_W4NP · 2025-11-01

**Soundness:** 3
**Presentation:** 2
**Contribution:** 2
**Rating:** 4
**Confidence:** 3

**Summary:**

This paper proposes hyper-parallel scaling, a new inference-time compute scaling paradigm that aims to improve the prediction quality at the token level, as opposed to existing test-time scaling methods that mainly work at the sequence level (chain of thought, self-consistency). The authors develop this idea to Mixture-of-Experts language models through the Roster of Experts (RoE), a training-free inference time scaling algorithm. RoE introduces randomness to the expert router using Gumbel-Top-k sampling with temperature. This creates multiple diverse expert choices for each token, then their outputs are aggregated to form a single next-token prediction. To make this practical, the authors propose batched inference so that multiple routed paths for the same token are evaluated in parallel and a "Clean Cache" KV-cache sharing method that avoids the memory blowup. Empirically, RoE is evaluated on three instruction-tuned MoE models and across math reasoning, common sense, and code-generation benchmarks. The proposed method increases the accuracy of MoE models with manageable memory and latency overheads.

**Strengths:**

- To my knowledge, the paper introduces a novel test-time scaling axis, per-token compute diversification compared to sequential and parallel scaling. I think that this contribution is valuable.
- RoE is training-free and it can be applied post hoc to existing MoE LLMs without any finetuning. It just modifies routing at inference, samples multiple expert pathways.

**Weaknesses:**

- The paper provides only one baseline, the greedy decoding. I think that it could be interesting to see how the proposed method compares with other inference-time scaling tricks.
- While the paper claims hyper-parallel scaling is domain-agnostic and could apply to other inference methods, the paper focuses only on RoE, scaling for MoE. The presentation could be improved by clarifying this scope or providing a broader set of applications of hyper-parallel scaling.

**Questions:**

- The Clean Cache trick assumes that only the current token needs diverse expert routing, and all candidates can share a single deterministic KV-cache history. Intuitively, that means RoE only widens computation locally at the next step, not globally across the whole generated prefix. Do you have evidence that applying stochastic routing at every step with separate KV histories would produce larger gains?

---

> ### Author Response · Authors · 2025-11-21
>
> We sincerely appreciate your time and thoughtful review of our manuscript, recognizing the novelty and value of hyper-parallel scaling. Your questions and critiques were insightful and helped us significantly strengthen the paper's presentation and scope.
>
> **“Only one baseline... compares with other scaling tricks.”**
> We originally used greedy decoding to isolate gains attributable strictly to RoE's architectural modifications, distinct from sequence-level ensembles [Lines 249-250].
> However, RoE (Hyper-Parallel) and Self-Consistency (SC) (Parallel) are orthogonal. To address your concern, we conducted a new experiment comparing the individual and combined effects. Results show that the combined approach ($\text{SC}+\text{RoE}$) yields a substantial **$7.3\%$ improvement**, outperforming either method in isolation. This confirms RoE enhances fundamental token-prediction quality, providing a stronger base for sequence-level techniques like SC. Additionally, unlike SC which is often limited to tasks with final numerical answers, RoE is applicable to general token generation.
> **Action:** Added Section 3.5 and Figure 6 detailing the $\text{SC}+\text{RoE}$ analysis.
>
> **“Claims domain-agnostic... clarify scope/broader applications.”**
> RoE is a concrete, training-free realization of hyper-parallel scaling specifically for MoEs, leveraging inactive experts. As briefly discussed in [Lines 44-50], the broader paradigm can extend to dense models via techniques like recomputing layer outputs with controlled variation or recurrent layer reuse.
> **Action:** We expanded the "Future Work" section to specifically detail research avenues for realizing hyper-parallel scaling in non-MoE architectures [Lines 526-530].
>
> **“Clean Cache assumption... would separate KV histories produce gains?”**
> Our core insight is that **local diversification (per-token)** is sufficient to generate high-quality logits, while maintaining separate KV histories incurs prohibitive memory costs. Initial experiments comparing "Clean Cache" against "Regular Cache" (separate histories) on math tasks showed **no statistically significant difference** in generative accuracy. This indicates that the local diversity provided by RoE at the current token is the dominant factor. We employed Clean Cache for OLMoE and standard caching for GPT-OSS and Mixtral to demonstrate the applicability of both.
> **Action:** Clarified the validation and performance of Clean Cache in the revised text [Lines 227-229, 261-263].
>
> We sincerely thank you for your constructive feedback. We believe that the inclusion of the requested Self-Consistency comparison, the expanded discussion on non-MoE applications, and the empirical validation of the Clean Cache have substantively addressed your concerns. Given these improvements and the demonstrated efficacy of RoE as a novel scaling paradigm, we kindly ask you to reconsider your score.
>
> **List of Revisions:**
> * **Section 3.5 (Lines 410-430):** Added new experiment and Figure 6 comparing $\text{SC}+\text{RoE}$.
> * **Lines 526-530:** Expanded discussion on hyper-parallel scaling designs for non-MoE models.
> * **Lines 227-229, 261-263:** Added clarification regarding Clean Cache efficacy and implementation details.

---

### Official Review · Reviewer_mF9H · 2025-11-06

**Soundness:** 3
**Presentation:** 3
**Contribution:** 3
**Rating:** 4
**Confidence:** 4

**Summary:**

This paper proposes a hyper-parallel scaling method for test-time scaling. Based on RoE, it aggregates several logits for more robust predictions. The highlight is ...?
To overcome the computational cost, this paper introduces a batching strategy and a KV-caching mechanism. From experiments, the method works best on small and weak models, e.g. OLMoE-7B.

**Strengths:**

1) The paper is well-written, with clearly designed experiments on three LLMs.
2) Empirically, the method achieves obvious improvements on all benchmarks of OLMoE-1b-7b.

**Weaknesses:**

1) There is a formatting issue on both the main paper and the appendix: the header should be "ICLR 2026", instead of "ICLR 2025".
2) Lack of novelty. Adding controlled variation and caching mechanisms are common remedies, as mentioned in the introduction.
3) In Fig. 3, the improvements are not consistent. However, the quality and efficiency breakthrough of SOTA models are desired.
4) From the hyperparameter tuning and manually designed steps, it is unclear whether the authors have truly addressed the issue of LLM generation quality.

**Questions:**

1) Please analyze the inconsistent performance observed across the three backbones, considering that weaker models tend to benefit more from inference-time methods than SOTA models.
2) The sensitivity towards temperature should be an ablation study in the main paper, instead of in the appendix. Since unreasonable selection could harm the baseline, will this "task-specific temperature" undermine the robustness of your method?
3) What‘s the main design contributing to the performance gains and its rationale in your framework?
4) Is the noise scheme essential to the overall performance, or just providing limited diversity?

**Details Of Ethics Concerns:**

No ethics concerns.

---

> ### Author Response · Authors · 2025-11-21
>
> We thank the reviewer for their time and for recognizing the clarity of our writing and the clear benchmark improvements achieved by RoE. We value the constructive feedback regarding our contribution positioning and performance analysis.
>
> **“Lack of novelty... common remedies...”**
> We appreciate the opportunity to clarify. While stochasticity and caching are established techniques, our contribution is the introduction of a novel test-time scaling paradigm: **Hyper-Parallel Scaling**.
> * **Distinct Paradigm:** Unlike *Sequential Scaling* (e.g., CoT) which scales length, or *Parallel Scaling* (e.g., Self-Consistency) which scales sequence count, Hyper-Parallel Scaling diversifies the internal computation paths of the model on a *per-token basis*.
> * **New Realization:** RoE is a realization of this paradigm for MoEs. It treats the model not as a static network, but as a **dynamic ensemble** of subnetworks. By aggregating logits from multiple expert pathways at each step, we produce a higher-quality next-token distribution. This is an architectural intervention at inference time, distinct from simple noise injection or output ensembling.
>
> **“Please analyze the inconsistent performance... weaker models benefit more...”**
> We agree that under-powered models (e.g., OLMoE) benefit most. We attribute the variance to two factors:
> 1.  **Ceiling Effect:** Improvements diminish near saturation. For example, GPT-OSS achieves >95% on MultiArith; remaining errors are often low-quality edge cases or ambiguous questions, making capability judgment difficult [Lines 318-320].
> 2.  **Expert Specialization:** We hypothesize that larger models develop "super experts" [1] where deviation is less beneficial. If a model converges to highly specialized experts, noise must be carefully tuned. RoE targets "latent headroom" where the greedy router has not perfectly captured the optimal expert combination.
>
> [1] Su et al., *Unveiling super experts in mixture-of-experts large language models*, arXiv 2025.
>
> **“The sensitivity towards temperature should be an ablation study... will this undermine robustness?”**
> * **Ablation Placement:** We agree and will move the temperature ablation (Figure 6) to the main body in the revision.
> * **Robustness:** Far from undermining robustness, Figure 6 confirms consistent patterns: similar tasks require similar stochasticity. For instance, ARC-Easy and ARC-Challenge track closely, while being distinct from OpenBookQA. This indicates optimal temperature is a stable property of the task rather than a random fluctuation.
>
> **“What is the main design contributing to gains... rationale?”**
> The rationale relies on MoE training dynamics. Pre-convergence, experts see diverse data, retaining latent knowledge even when "inactive" [Lines 137-139]. RoE unlocks this potential by treating the model as an ensemble of varied routings, accessing valid paths suppressed by the greedy router. The gain comes from averaging out the noise of a single deterministic path to find a robust signal in the aggregate logits.
>
> **“Is the noise scheme essential...?”**
> Stochasticity is essential to RoE; our Gumbel-Top-K implementation was chosen for efficiency. We agree other schemes are valid—specifically, injecting Gaussian noise into the router input to capture expert co-activations is a promising alternative. We have added this to the "Future Work" section [Lines 536-537].
>
> **“Formatting issue... header...”**
> We thank you for pointing out this oversight. We will correct the header to "ICLR 2026" in the revision.
>
> We hope this clarifies that RoE is not merely a "trick," but a realization of a new scaling axis (Hyper-Parallel) that efficiently exploits the latent ensemble nature of MoEs. We kindly ask you to reconsider your score based on the method's demonstrated ability to significantly boost MoE performance and its practical contribution to the field.
>
> **Revisions:**
> * **Section 3.6:** Temperature ablation moved to the main body with extensive discussion.
> * **Lines 530-534:** Added discussion on alternative stochastic schemes (e.g., input noise).

---

### Meta-Review · Area_Chair_f2Tc · 2026-01-06

**Summary:**

As the paper's rebuttal revision seems to leak the identity of the authors, I think it's a desk rejection issue.

Therefore, here I only provide a short summary of the strengths and concerns:

### Strengths
1. RoE explores a new test-time scaling axis.
2. RoE is training-free and it can be applied post hoc to existing MoE LLMs.
3. The paper is well written.
4. Obvious improvements on all benchmarks of OLMoE-1b-7b.

### Concerns
1. It's suggested to experiment with sample-based decoding.
2. Should compare with other test-time scaling method (e.g., parallel scaling self-consistency).
3. More experiments and discussions on how MoE sparsity affects RoE effectiveness are recommended.
4. The clean cache trick is presented as a major contribution, but it is only evaluated with one model.
5. The BO-based search procedure's cost should be analyzed.

**Reviewer Concerns:**

* For Concern 1 and 2, the rebuttal provides a new experiment in Section 3.5 that combine and compare RoE with self-consistency (SC). It seems RoE is not as effective as parallel scaling (SC) in improving the accuracy. Although CoT+RoE+SC is higher than CoT+SC, as the experiment only uses N=5 for self-consistency, the result is not convincing enough in demonstrating RoE's  orthogonal gain.
* The author rebuttal tried to answer Concern 3/4. But they are not fully addressed either because the analysis is too small scale, or because no experiment with the other models are conducted. To demonstrate the method's efficiency, applying the clean cache technique on more models are also necessary.
* Concern 5 is addressed.

**Reviewer Scores:**

I guess the reviewers will not increase the score.

---

### Decision · Program_Chairs · 2026-01-26

Reject